# Use of the Proboscis Extension Response Assay to Evaluate the Mechanism of House Fly Behavioral Resistance to Imidacloprid

**DOI:** 10.3390/insects15030168

**Published:** 2024-03-01

**Authors:** Sara D’Arco, Lara Maistrello, Caleb B. Hubbard, Amy C. Murillo, Alec C. Gerry

**Affiliations:** 1Interdepartmental Center BIOGEST-SITEIA, Department of Life Sciences, University of Modena and Reggio Emilia, 42122 Reggio Emilia, Italy; 283873@studenti.unimore.it (S.D.); lara.maistrello@unimore.it (L.M.); 2Department of Entomology, University of California, Riverside, CA 92521, USA; chubb001@ucr.edu (C.B.H.); amy.murillo@ucr.edu (A.C.M.)

**Keywords:** *Musca domestica*, insecticide, chemoreception, discrimination, aversion

## Abstract

**Simple Summary:**

This study investigated the detection and discrimination of the neonicotinoid insecticide imidacloprid by behaviorally resistant and susceptible house flies (*Musca domestica* L.). Flies were allowed to contact a sucrose solution containing either a low or a high concentration of imidacloprid with their tarsi alone or with both their tarsi and proboscis. The proboscis extension response (PER) for each house fly was recorded at 0, 2, and 10 s following the start of tarsal contact with the test solution. Following proboscis contact with the sucrose solution containing a high concentration of imidacloprid, behaviorally resistant flies had a significant reduction in PER (within 2 s), while imidacloprid-susceptible flies showed no differences in PER associated with the concentration of imidacloprid. When only the tarsi were allowed to contact either solution, there were no significant differences in PER observed for either fly strain (resistant or susceptible). These results suggest that behaviorally resistant house flies detect imidacloprid and can discriminate among low and high concentrations following proboscis contact but not tarsal contact with a sucrose solution containing imidacloprid. Understanding the mechanisms responsible for behavioral resistance to insecticides by the house fly is critical for creating sustainable pest management strategies for this fly.

**Abstract:**

The house fly, *Musca domestica* L., is a significant human and livestock pest. Experiments used female adult house flies glued onto toothpicks for controlled exposure of their tarsi alone (tarsal assay) or their tarsi and proboscis (proboscis assay) with a sucrose solution containing imidacloprid at either a low (10 µg/mL) or high (4000 µg/mL) concentration. Proboscis extension response (PER) assays were used to characterize the response of imidacloprid-susceptible and behaviorally resistant house fly strains to contact with sucrose solutions containing either a low or high concentration of imidacloprid. In each assay, 150 female flies from each fly strain were individually exposed to sucrose solutions containing either a low or high concentration of imidacloprid by deliberate contact of the fly tarsi to the test solution. The PER for each fly was subsequently recorded at 0, 2, and 10 s following the initial tarsal contact. A significant and rapid reduction in PER was observed only for the behaviorally resistant fly strain and only following contact by the flies’ proboscis with the sucrose solution containing the high imidacloprid concentration. The results suggest that chemoreceptors on the fly labellum or internally on the pharyngeal taste organs are involved in the detection of imidacloprid and discrimination of the concentration, resulting in an avoidance behavior (proboscis retraction) only when imidacloprid is at sufficient concentration. Further research is needed to identify the specific receptor(s) responsible for imidacloprid detection.

## 1. Introduction

The house fly (*Musca domestica* L.) is a major pest in confined animal facilities and is a potential mechanical vector of over 200 pathogens [1,2]. However, control of this pest has been complicated by the development of insecticide resistance to all major available insecticidal classes, including pyrethroids, organophosphates, carbamates, and neonicotinoids [3]. Insecticides formulated for controlling house flies are often applied as a component of sugar-based food bait (“fly bait”). Fly baits containing the neonicotinoid insecticide imidacloprid have been commercially available since 2002, and initially provided good control of house flies resistant to other available insecticides [4]. However, as the use of imidacloprid-containing bait for fly control increased, house fly resistance to imidacloprid was soon reported [5,6]. Gerry and Zhang (2009) suggested that house fly resistance to imidacloprid was due to the altered behavior of house flies following detection of or contact with imidacloprid [5]. Later studies confirmed that house flies express “behavioral resistance” by reduced feeding on the bait [7]. Genetic studies using the F_1_ backcross method [8] demonstrated that factors contributing to the behavioral resistance to imidacloprid were found on autosomes 1 and 4 of the house fly [9].

For “food-based” insecticides such as fly baits, consumption of the food material is required to achieve fly mortality, and thus the mechanisms of “taste” that determine the suitability or palatability of the bait to the fly are important to understand. In flies and other insects, tasting occurs through chemical detection by chemosensory taste receptors (gustatory receptors (GRs)) located within the taste sensilla present on the mouthparts, legs, antennae, ovipositor or even the wings [10,11]. In *Drosophila melanogaster* (Meigen), the taste sensilla include up to four molecularly and physiologically distinct taste neurons that are selectively activated by palatable (sweet, salty, water) or noxious (high salt, bitter, low pH) tastants [12]. In Diptera, taste receptors are most reported on the legs (particularly on the fore tarsi) [13], the labellum at the tip of the proboscis, and the pharyngeal organs lining the esophagus [12,14]. As flies land on or walk across a surface, the tarsi are often the first body structures to contact a potential food source. When taste receptors on the tarsi are stimulated by food molecules such as sugars, flies initiate an appetitive behavior sequence by extending their proboscis to contact the potential food source to further assess food quality and to initiate feeding if the food is deemed suitable [11]. Thus, proboscis extension begins the feeding process but follows the initial detection of a potential food source by taste receptors on the legs or other body structures [15,16].

House flies are reported to exhibit aversive behaviors to sucrose formulated with imidacloprid only following direct contact with the bait [7], suggesting that flies avoid imidacloprid only following the detection of imidacloprid by gustatory receptors rather than following detection of volatiles by odorant receptors. However, the specific mechanism for imidacloprid detection and the initiation of a behavioral response by the house fly is not well understood. The present study aimed to achieve two objectives. The first was to determine whether the detection of imidacloprid and discrimination of imidacloprid concentration by behaviorally resistant house flies occurs via the tarsi or the proboscis (labellum) and/or pharyngeal organs lining the esophagus. The second was to determine if susceptible and resistant fly strains varied in their ability to discriminate and respond to imidacloprid at low or high concentrations.

## 2. Materials and Methods

### 2.1. House Fly Strains

Two house fly strains (UCR, BRS) were used in this study. The UCR strain is an insecticide-susceptible strain maintained in the laboratory at UC Riverside since 1982 following the collection of pupae from a dairy farm in Mira Loma, California. The BRS strain was selected to exhibit a strong behavioral resistance phenotype to the insecticide imidacloprid [7] following collection via sweep net from a dairy farm in San Jacinto, CA. Following initial selection for the behavioral resistance phenotype, the BRS fly strain has been maintained under continuous selection pressure (selection every three filial generations) to maintain the selected level of behavioral resistance to imidacloprid. Both populations were maintained in insectary rooms under standard environmental conditions (27 °C, 14:10 L:D, and 35% RH) and reared following standard practices [17].

### 2.2. Proboscis Extension Response—Proboscis Contact

Proboscis extension response assays were conducted following the methods described by Shiraiwa and Carlson [18] with the following modifications: adult house flies (3–6 d old) were aspirated from an adult colony cage, placed into plastic holding cages without food (only water), and held in a standard laboratory room (at 22 ± 2 °C) for a starvation period of 1 d for the UCR strain and 2–3 d for the BRS strain. The length of starvation for each cohort of flies was determined by removing a small number of flies (~10–15) from the holding cage each day to test for a proboscis extension response (PER) to a 30% sucrose solution (as described below). When >60% of tested flies showed a positive PER to the sucrose solution, flies in the cohort were deemed to be sufficiently starved for testing. Following the starvation period, flies in the holding cage were chilled in a −20 °C freezer to immobilize them for a few minutes and then sorted by sex on an electronic chill table (Catalog #1431, BioQuip Products Inc., Compton, CA, USA). Female flies were glued to flat wooden toothpicks by placing a drop of fast-drying clear nail polish (Seche Vite, Item 83100, American International Industries, Los Angeles, CA, USA) to the flat end of the toothpick and carefully touching the nail polish to the dorsal thorax of the fly (Figure 1). The glue holding the fly to the wooden toothpick was allowed to dry by placing the narrow end of the toothpick into a microtube rack well so that the flat end of the toothpick with the attached fly was suspended in the air, preventing the fly from contacting any surface. Glued flies were held in this position for at least 30 min to ensure full recovery from cold knockdown before their use in the feeding assays below.

Glued flies were initially tested for an appropriate PER following contact of fly tarsi with first a negative control (deionized water only) and then a positive control (30% sucrose solution) [18]. Each control solution (~1 mL) was pipetted onto a separate clean glass microscope slide, and the flies were manually manipulated so that the tarsi contacted the control solution. Flies exhibiting a PER to the positive control solution were allowed to contact the positive control solution with their proboscis for up to 2 s before the fly was withdrawn. Flies exhibiting the appropriate PER to each control solution were subsequently used in additional assays (as described below) after first rinsing the fly tarsi in deionized water and then dabbing the tarsi on clean tissue to remove any remaining liquid residue.

Flies were subsequently examined for PER to test solutions of 30% sucrose containing imidacloprid at either a low concentration (10 µg/mL) or a high concentration (4000 µg/mL). The low concentration was selected on the basis of findings by Hubbard and Murillo [19] that flies from the behaviorally resistant BRS strain readily feed and survive on granular sucrose containing imidacloprid at concentrations of <100 µg/g. Thus, the low imidacloprid concentration was not expected to result in an aversive behavior in these resistant flies. The high concentration of 4000 µg/mL imidacloprid (Chem Service, Inc. West Chester, PA, USA) was used because this was the challenge concentration Hubbard and Gerry (2020) used for selecting and maintaining behavioral resistance in the BRS fly strain, and these flies showed a significant reduction in feeding on imidacloprid at this concentration [7,19].

The flies were held so that the tarsi contacted the test solution and the flies could also reach the test solution by extending their proboscis. Flies extending their proboscis immediately following tarsal contact with the test solution were allowed to maintain continuous proboscis contact with the test solution for up to 10 s. The presence or absence of PER was observed and recorded at 0, 2, and 10 s of continuous proboscis contact with the test solution. Any fly retracting the proboscis between these observation times was recorded as PER- at the next observation time, and testing was concluded for that individual fly. Five replicate cohorts of 30 flies each (*n* = 150 flies) were tested for each house fly strain (UCR, BRS) and for each concentration of imidacloprid (low, high) for a total of 600 flies tested.

### 2.3. Proboscis Extension Response—Tarsal Contact

Adult house flies were aspirated from adult colony cages and handled as described for the proboscis contact assay above, except that flies were held in a position that allowed the flies to contact the control and test solutions with their tarsi only and prevented contact with their proboscis. As previously described in the proboscis assay, flies responding appropriately to both the negative and positive control solutions were examined for PER to test solutions formulated with 30% sucrose and imidacloprid at either a low concentration (10 µg/mL) or a high concentration (4000 µg/mL). In this tarsal contact assay, the flies were unable to reach the test solution with their extended proboscis. Flies extending their proboscis immediately following tarsal contact with the test solution were allowed to exhibit PER without proboscis contact with the test solution for up to 10 s. The presence or absence of PER was observed and recorded at 0, 2, and 10 s of continuous proboscis extension. Any fly retracting the proboscis between observation times was recorded as PER- (not exhibiting PER) at the next observation time, and testing was concluded for that individual fly. Five replicate cohorts of 30 flies each (*n* = 150 flies) were tested for each house fly strain (UCR, BRS) and for each concentration of imidacloprid (low, high) for a total of 600 flies tested.

### 2.4. Statistical Analysis

Data were analyzed separately for each assay (proboscis or tarsal contact) and fly strain (BRS, UCR). The number of flies exhibiting PER in each replicate group was rank ordered within each observation time (0, 2, and 10 s) and then analyzed using Friedman’s test (non-parametric ranked ANOVA) to test for an overall difference in PER among the two concentrations of imidacloprid and for significant interactions between the observation time and the concentration of imidacloprid. Rank values were further analyzed within each observation time by Wilcoxon’s rank sum test for differences in the number of flies exhibiting PER for the two concentrations of imidacloprid at the same observation timepoint, with the significance adjusted for multiple comparisons within each assay (α = 0.016). All statistics were performed in R v 4.3.1 [20].

## 3. Results

The mean proboscis extension response (PER) by fly strain and observation time is provided in Table 1 (proboscis contact assay) and Table 2 (tarsal contact assay). The mean number of flies exhibiting PER predictably decreased across sequential observation times (0, 2, and 10 s) for all cohort groups, regardless of the assay method, fly strain, or concentration of imidacloprid, as flies that retracted their proboscis at any point in the assay were removed from the assay before the next observation time.

### 3.1. Proboscis Contact Assay

Immediately upon tarsal contact with sucrose containing imidacloprid at the 0 s observation time (prior to the first contact by the proboscis), a similar mean number of flies extended their proboscis to both the low and high concentration of imidacloprid for the imidacloprid-susceptible UCR flies (14.0 ± 1.18 and 15.6 ± 0.98, respectively) and the imidacloprid-resistant BRS flies (18.0 ± 0.71 and 18.4 ± 0.51, respectively) (Table 1).

Across all observation times, the concentration of imidacloprid had no effect on the number of UCR flies exhibiting PER (F = 0.28; df = 1,29; *p* = 0.60), and there was no interaction between observation time and the concentration of imidacloprid (F = 0.80; df = 2,29; *p* = 0.37). Similarly, within each observation time, there was no difference in the number of UCR flies exhibiting PER between the concentrations of imidacloprid at 0 s (W = 18, *p* = 0.27), 2 s (W = 13, *p* = 0.98), or 10 s (W = 10, *p* = 0.66) (Figure 2A). Thus, the proportion of UCR flies exhibiting PER decreased similarly across subsequent observation times for both the low and high concentrations of imidacloprid, with 8.8 ± 1.28 and 9.0 ± 1.38 of the flies, respectively (~30% for each group), continuing to exhibit PER at 2 s, followed by 5.4 ± 0.75 (18%) and 4.6 ± 0.81 (15%) of the flies, respectively, still exhibiting PER at 10 s. After adjusting the PER for flies removed from the assay at each previous observation time (Figure 3A), ~50–60% of the UCR flies were noted to have continued exhibiting PER from one observation time to the next, with no significant difference between the low or high concentrations of imidacloprid (W > 9, *p* > 0.05).

In contrast to the UCR flies, the concentration of imidacloprid had a significant effect on the number of BRS flies exhibiting PER (F = 14.79; df = 1,29; *p* = 0.0007), and there was also a significant interaction between the observation time and the concentration of imidacloprid (F = 5.81; df = 2,29; *p* = 0.008). Within the observation times, the number of flies exhibiting PER did not vary with the concentration of imidacloprid at 0 s (W = 14.5, *p* = 0.8), but there was a significant difference at 2 s (W = 0, *p* = 0.007) and 10 s (W = 0, *p* = 0.006) (Figure 2B). At the 2 s observation time, 14.2 ± 0.97 (45%) of BRS flies continued to exhibit PER to the low concentration of imidacloprid, while only 6.4 ± 0.68 (22.5%) of BRS flies continued to exhibit PER to the high concentration of imidacloprid. This difference carried over to the 10 s observation time, with 10.0 ± 1.3 (35%) of BRS flies exhibiting PER to the low concentration of imidacloprid, while only 3.2 ± 0.58 (12%) of BRS flies exhibited PER to the high concentration of imidacloprid. After adjusting the PER for flies removed from the assay at each previous observation time, it was clear that the significant reduction in PER occurred only at the 2 s observation time (W = 0, *p* = 0.007), when PER continued for 79% of the BRS flies exposed to the low concentration of imidacloprid but only 35% of the BRS flies exposed to the high concentration of imidacloprid (Figure 3B). While the adjusted PER was also lower at 10 s for BRS flies exposed to the high concentration of imidacloprid (49%) relative to the low concentration of imidacloprid (70%), this difference was not significant (W = 3, *p* = 0.06).

### 3.2. Tarsal Contact Assay

Upon initial tarsal contact with sucrose containing imidacloprid at the 0 s observation time, a similar number of flies exhibited PER to both the low and high concentrations of imidacloprid for the imidacloprid-susceptible UCR flies (13.0 ± 0.77 and 13.8 ± 1.43, respectively) and the imidacloprid-resistant BRS flies (16.0 ± 1.18 and 12.4 ± 2.25, respectively) (Table 2). These mean PER values were similar to those observed for both fly strains in the proboscis contact assay at 0 s before the flies were allowed to contact the test solutions using their proboscis.

Across all observation times in the tarsal contact assay, the concentration of imidacloprid had no effect on the number of UCR flies exhibiting PER (F = 0.62; df = 1,29; *p* = 0.43), and there was no interaction between the observation time and the concentration of imidacloprid (F = 1.14; df = 1,29; *p* = 0.29). Within each observation time, there also was no difference in the number of UCR flies exhibiting PER between the concentrations of imidacloprid at 0 s (W = 13, *p* = 0.95), 2 s (W = 9, *p* = 0.5), or 10 s (W = 12.5, *p* = 1) (Figure 4A). Although there was an overall effect of the concentration of imidacloprid on the number of BRS flies exhibiting PER (F = 12.89; df = 1,29; *p* = 0.001), there was no interaction between the observation time and the concentration of imidacloprid (F = 0.49; df = 1,29; *p* = 0.52) and there were no differences in PER by the concentration of imidacloprid within each observation time at 0 s (W = 6.5, *p* = 0.26), 2 s (W = 4, *p* = 0.09), or 10 s (W = 2.5, *p* = 0.04) when the *p*-values were adjusted for multiple comparisons (Figure 4B).

For both the UCR and BRS fly strains, the proportion of flies exhibiting PER in the tarsal contact assay decreased similarly across subsequent observation times for both the low and high concentrations of imidacloprid. For UCR flies exposed to the low- or high-imidacloprid solution, 10.0 ± 1.45 (33%) or 9.4 ± 0.81 (31%) of the flies, respectively, continued to exhibit PER at 2 s, followed by 5.4 ± 1.72 (18%) or 5.6 ± 1.36 (19%) still exhibiting PER at 10 s (Figure 4A). For BRS flies exposed to the low- or high-imidacloprid solution, 10.0 ± 1.30 (33%) or 6.4 ± 1.21 (21%) of the flies, respectively, continued to exhibit PER at 2 s, followed by 7.2 ± 0.73 (24%) or 4.0 ± 0.89 (13%) still exhibiting PER at 10 s (Figure 4B). After adjusting PER for flies removed from the assay at each previous observation time, a similar proportion of UCR flies and BRS flies continued to exhibit PER at 2 s (70–75% and 52–62%, respectively) and at 10 s (50–60% and 65–74%, respectively) (Figure 5A,B), with no differences between concentrations of imidacloprid at any observation time (W > 2.5, *p* > 0.016).

## 4. Discussion

House flies from both fly strains exhibited PER when placed in tarsal contact with the sucrose solutions, as would be expected, given previous studies demonstrating the detection of sugars by the tarsi [21,22,23,24]. To understand taste detection and the associated behaviors, it is important to differentiate between detection and discrimination. Detection refers to identifying the presence of a substance, while discrimination refers to the ability to distinguish between different concentrations of the substance. According to our results, there was no evidence of the detection of imidacloprid or at least discrimination between the low (non-lethal) or high (lethal) concentrations of imidacloprid by the resistant BRS flies when the flies contacted the sucrose solutions using their tarsi alone. Instead, the BRS flies exhibited a variable response to the solutions with low or high concentrations of imidacloprid only following proboscis contact with the solutions. Thus, the behavioral resistance to imidacloprid (reduced feeding) previously reported for these flies [7] most likely results from activation of the gustatory receptors (GRs) associated with the labellum or other mouthparts following contact with imidacloprid. This was supported by the rapidity of the response, with substantial numbers of resistant BRS flies quickly retracting their proboscis within 2 s of initial proboscis contact with the high-imidacloprid solution. A rapid retraction of the proboscis would limit exposure to and especially uptake (by feeding) of the toxic food, an important consideration, given that the behaviorally resistant BRS flies lack substantial physiological resistance to imidacloprid [7] and thus would die if a high dose of imidacloprid was consumed in more than trace amounts. In contrast, the imidacloprid-susceptible UCR flies continued to exhibit PER similarly to both the low- and high-imidacloprid solutions during the proboscis contact assay when the flies were allowed to contact the solutions with their proboscis for up to 10 s.

The BRS house fly strain used in this study was previously selected for a high level of behavioral resistance to imidacloprid, which resulted in the BRS flies significantly reducing their contact time with sucrose treated with imidacloprid relative to sucrose alone [7,9]. The similar landing rate of BRS house flies on sucrose with or without imidacloprid [7] suggests that these behaviorally resistant flies cannot detect imidacloprid prior to physical contact of their proboscis with an imidacloprid-treated sugar source. Furthermore, the selected resistance was specific to imidacloprid, as BRS flies readily consumed sugar baits containing another related neonicotinoid insecticide (dinotefuran) [7].

The gustatory system is responsible for detecting non-volatile cues in the environment and is primarily involved in feeding behavior, allowing animals to detect and discriminate between nutritious and noxious foods [25]. Insects have gustatory receptor neurons (GRNs) that are widely distributed over the body surface, and activation of the GRNs in different peripheral tissues will mediate distinct behaviors [25,26]. A highly conserved clade of GRs plays a critical role in the detection of and response to chemical compounds as part of the insect taste system (e.g., [27,28]). In Diptera, taste detection is mediated by sensory bristles on the proboscis, internal mouthparts, legs, wings, and ovipositor [25,29], with activation of the gustatory neurons on the tarsal leg segments following contact with sugars resulting in proboscis extension and the initiation of feeding [25,26]. The taste organs of house flies are predominantly located on the labellum at the tip of the proboscis and on the tarsi [18,30]. Taste receptors on the legs are common among many insects, including Lepidoptera [31], Hymenoptera [32], Orthoptera [33], Coleoptera [34], and Blattodea [11,35].

Georghiou (1972) categorized behavioral resistance as either stimulus-independent or stimulus-dependent. Stimulus-independent behavioral resistance is a result of the insect’s natural avoidance of an environment or situation where it might be exposed to an insecticide [7,36]. Stimulus-dependent behavioral resistance refers to an insect’s increased ability to detect and limit contact with a toxic substance, possibly due to the substance’s repellent or irritant properties, formulation, or presentation, resulting in an aversive response [7,36]. In our study, the flies were unable to avoid tarsal contact with the sucrose solution containing imidacloprid due to the design of the assay, but they could avoid extending their proboscis or they could retract their proboscis during the trial period, making their proboscis extension response (PER) dependent on the detection and discrimination of the concentration of imidacloprid.

In general, insects can use multiple mechanisms to avoid consuming a toxicant present in a sugar food bait including (1) activation of bitter-sensing GRNs by a bait component resulting in the cessation of feeding or (2) inhibition of sugar-sensing GRNs, reducing recognition of the sugar bait as a suitable food source [37,38]. *Drosophila* spp. discriminate among sugar concentrations using GRNs on the tarsi and can be trained to avoid sugar concentrations when these are associated with a negative stimulus [39]. An unusual gain-of-function adaptation in some populations of the German cockroach (*Blattella germanica* L.) provides resistant cockroaches with protection from toxic baits containing glucose as the result of an acquired sensitivity to glucose with both the activation of bitter-sensing GRNs and the suppression of sugar-sensing GRNs following contact with glucose [40,41,42].

A study found that bait-resistant cockroach strains also discriminate between different concentrations of glucose, with an inverse relationship between the concentration of glucose and feeding response when aqueous solutions of glucose were presented to the cockroaches’ paraglossae following ablation of the maxillary and labial palps [40]. According to our results, discrimination between the low and high concentrations of imidacloprid by the resistant BRS fly strain occurred after proboscis contact with the test solutions; however, the specific location of the GRNs associated with the house fly’s mouthparts and the specific GRs responsible for the detection of imidacloprid by these house flies are not known. In *Drosophila*, bitter-sensing GRNs that detect aversive tastants, including noxious substances, are characterized by subsets of GRs that do not overlap with those expressed in sweet-sensing GRNs [43,44]. The rapid proboscis retraction of BRS flies after proboscis contact with the high-imidacloprid solution suggests a strong aversive behavior, perhaps due to selection in these flies for the dose-dependent activation of bitter-sensing GRNs by imidacloprid, allowing for greater discrimination of the concentration of imidacloprid to avoid a lethal exposure to this toxicant.

Investigating insecticide resistance in field populations can provide insights into evolutionary processes. Strong selective agents and pressure can lead to the rapid evolution of resistance. In some cases, behavioral resistance to an insecticide can provide greater protection than physiological resistance, since resistance cannot be overcome by increasing the concentration of insecticide [9]. Additionally, behavioral resistance has been shown to be stable over time, even in the absence of exposure to imidacloprid, suggesting that implementing traditional insecticide resistance management approaches, such as rotating or temporarily halting the use of an insecticide, may not be effective in reducing behavioral resistance [45]. House fly susceptibility to imidacloprid was high soon after the release of the first commercial fly bait containing this insecticide [4], but the bait’s effectiveness quickly deteriorated, likely due to rapid selection for behavioral resistance to imidacloprid in house fly populations under intense selection pressure [46,47]. Given the specificity of BRS flies for behavioral resistance to imidacloprid relative to the related neonicotinoid dinotefuran [7], it seems that either the GRs specifically detect imidacloprid, or if dinotefuran is detected by fly mouthpart-associated GRs, these flies are unable to discriminate a lethal concentration of this toxicant. Future studies to characterize the specific GRs involved in the detection of imidacloprid could guide the structural modification of imidacloprid to avoid its detection or discrimination by resistant flies, thereby rescuing the imidacloprid compound as a useful toxicant for fly control.

The PER assay as described in this study would be a useful way to evaluate the progression of behavioral resistance resulting from selection for taste aversion. These assays also provide greater detail on the mechanisms of resistance than typical insecticide exposure and mortality assays. For example, the rate of proboscis retraction could provide clues as to how the toxicant is detected, and the presence or absence of PER to different concentrations of a toxicant could provide insights into dose discrimination or substance detection thresholds.

## Figures and Tables

**Figure 1 insects-15-00168-f001:**
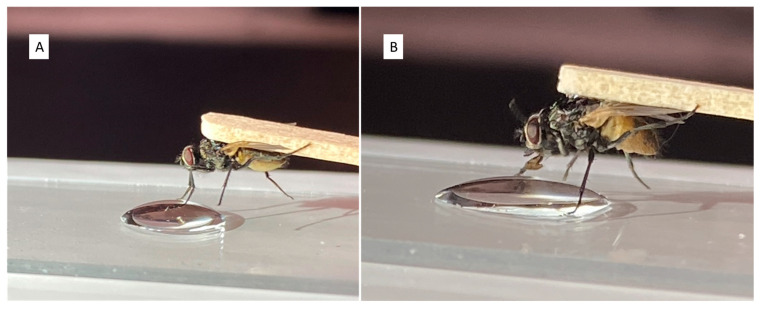
House flies glued on their dorsal thorax to the flat end of a wooden toothpick for PER assays. (**A**) A fly failing to exhibit PER following tarsal contact with a water control. (**B**) A fly demonstrating PER following tarsal contact with the sucrose control.

**Figure 2 insects-15-00168-f002:**
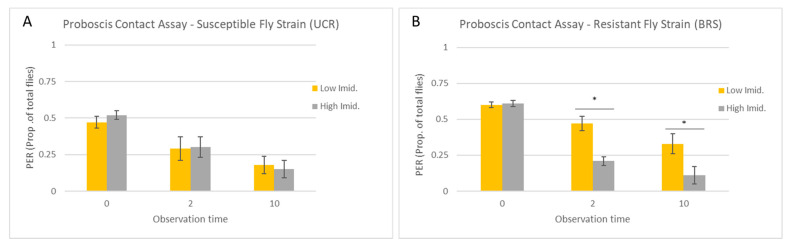
Proboscis contact assay. Flies were allowed to contact the solution with both the tarsi and proboscis. Columns show the mean proportion of UCR-susceptible (**A**) or BRS-resistant (**B**) house flies (five replicate groups of 30 flies = 150 flies per fly strain) that exhibited a continuous proboscis extension response (PER) at 0, 2, and 10 s following the start of tarsal contact with a sucrose solution containing imidacloprid at either a low (10 µg/mL) or high (4000 µg/mL) concentration (error bars indicate the standard error of the mean). The number of flies exhibiting PER was analyzed using Wilcoxon’s rank sum test with the *p*-value modified for multiple comparisons (*p* < 0.016) and with differences between the concentrations imidacloprid indicated by an asterisk (*).

**Figure 3 insects-15-00168-f003:**
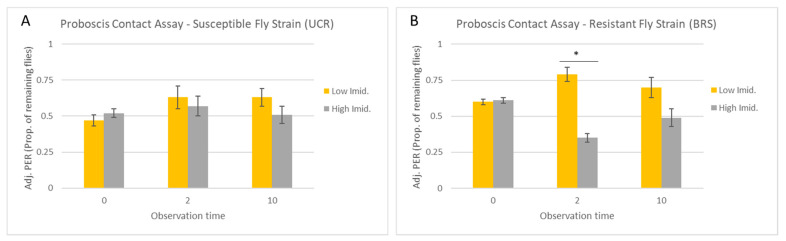
Proboscis contact assay: adjusted PER. Flies were allowed to contact the solution with both the tarsi and proboscis. Columns show the proportion of flies remaining from the previous observation time that continued to exhibit PER. Flies removed at each timepoint were thus not included in the PER calculation for the next timepoint. Columns show the adjusted PER for UCR-susceptible (**A**) and BRS-resistant (**B**) house flies following contact with a sucrose solution containing imidacloprid at either a low (10 µg/mL) or high (4000 µg/mL) concentration (error bars indicate the standard error of the mean). The proportion of remaining flies exhibiting PER was analyzed using Wilcoxon’s rank sum test, with the *p*-value modified for multiple comparisons (*p* < 0.016) and with differences between concentrations of imidacloprid indicated by an asterisk (*).

**Figure 4 insects-15-00168-f004:**
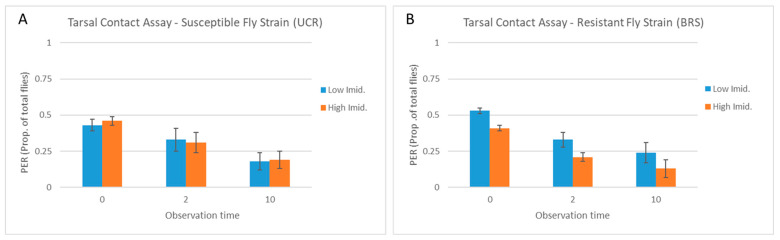
Tarsal contact assay. Flies were allowed to contact the solution with the tarsi only. Columns show the mean proportion of UCR-susceptible (**A**) and BRS-resistant (**B**) strain house flies (five replicate groups of 30 flies = 150 flies per fly strain) that exhibited a continuous proboscis extension response (PER) at 0, 2, and 10 s following the start of tarsal contact with a sucrose solution containing imidacloprid at either a low (10 µg/mL) or high (4000 µg/mL) concentration (error bars indicate the standard error of the mean). The number of flies exhibiting PER was analyzed using Wilcoxon’s rank sum test with the *p*-value modified for multiple comparisons (*p* < 0.016). There were no significant differences between concentrations of imidacloprid within observation times.

**Figure 5 insects-15-00168-f005:**
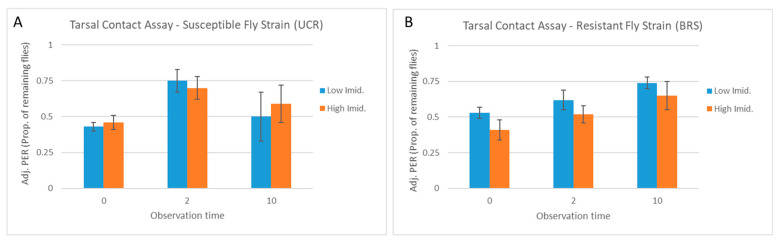
Tarsal contact assay: adjusted PER. Flies were allowed to contact the solution with the tarsi only. Columns show the proportion of flies remaining from the previous observation time that continued to exhibit PER. Flies removed at each timepoint were thus not included in the PER calculation for the next timepoint. Columns show the adjusted PER for UCR-susceptible (**A**) and BRS-resistant (**B**) house flies following contact with a sucrose solution containing imidacloprid at either a low (10 µg/mL) or high (4000 µg/mL) concentration (error bars indicate the standard error of the mean). The proportion of remaining flies exhibiting PER was analyzed using Wilcoxon’s rank sum test, with the *p*-value modified for multiple comparisons (*p* < 0.016). There were no significant differences between concentrations of imidacloprid within observation times.

**Table 1 insects-15-00168-t001:** Proboscis extension response (PER) of susceptible and resistant house flies (*n* = 30 flies/replicate; five replicates) during the proboscis contact assay observed at three times (0, 2, 10 s) following initial tarsal contact with a sucrose solution containing imidacloprid at either a low (10 µg/mL) or a high (4000 µg/mL) concentration. Bold values indicate significant differences in PER between high and low concentrations of imidacloprid determined using Wilcoxon’s rank sum test with the *p*-value adjusted for multiple comparisons within each assay (α = 0.016).

Trial	Strain	Imidacloprid Concentration	Observation Time (s)	PER (Mean + SE)	*p* Value	W
Proboscis assay	UCR (susceptible)	High	0	15.60 ± 0.98	0.27	18
		Low		14.00 ± 1.18		
		High	2	9.00 ± 1.38	0.98	13
		Low		8.80 ± 1.28		
		High	10	4.60 ± 0.81	0.66	10
		Low		5.40 ± 0.75		
	BRS (resistant)	High	0	18.40 ± 0.51	0.8	14.5
		Low		18.00 ± 0.71		
		High	2	**6.40 ± 0.68**	**0.007**	**0**
		Low		**14.20 ± 0.97**		
		High	10	**3.20 ± 0.58**	**0.006**	**0**
		Low		**10.00 ± 1.30**		

**Table 2 insects-15-00168-t002:** Proboscis extension response (PER) of susceptible and resistant house flies (*n* = 30 flies/replicate; five replicates) during the tarsal contact assay observed at three times (0, 2, 10 s) following initial tarsal contact with a sucrose solution containing imidacloprid at either a low (10 µg/mL) or a high (4000 µg/mL) concentration. The analyses were performed using Wilcoxon’s rank sum test with the *p*-value adjusted for multiple comparisons within each assay (α = 0.016).

Trial	Strain	Imidacloprid Concentration	Observation Time (s)	PER (Mean + SE)	*p* Value	W
Tarsal assay	UCR (susceptible)	High	0	13.8 ± 1.43	0.95	13
		Low		13.0 ± 0.77		
		High	2	9.4 ± 0.81	0.5	9
		Low		10.0 ± 1.45		
		High	10	5.6 ± 1.36	1	12.5
		Low		5.4 ± 1.72		
	BRS (resistant)	High	0	12.4 ± 2.25	0.26	6.5
		Low		16.0 ± 1.18		
		High	2	6.4 ± 1.21	0.09	4
		Low		10.0 ± 1.30		
		High	10	4.0 ± 0.89	0.04	2.5
		Low		7.2 ± 0.73		

## Data Availability

The datasets used and/or analyzed during the current study are available from the corresponding author on reasonable request.

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
