# Peer review of "Use of the Proboscis Extension Response Assay to Evaluate the Mechanism of House Fly Behavioral Resistance to Imidacloprid"

_insects, 2024, doi:10.3390/insects15030168_

Round 1

Reviewer 1 Report

Comments and Suggestions for Authors

insects-2863337

Title: Use of the Proboscis Extension Response Assay to Evaluate the Mechanism of House Fly Behavioral Resistance to Imidacloprid

Brief.

The manuscript presents a study to investigate delves into the ability of behaviorally resistant and susceptible house flies (Musca domestica L.) to detect or discriminate the neonicotinoid insecticide imidacloprid. Through controlled exposure experiments, flies interacted with sucrose solutions at varying concentrations using their tarsi alone or both tarsi and proboscis, shedding light on their distinct responses to low and high imidacloprid concentrations. The manuscript was well written, and the methodology approach is appropriate. In the general comments was left minor suggestions to improve the manuscript.

General comments

L80 - Please, the term 'dose' should be changed to 'concentration' in the sentence 'The first is to determine whether imidacloprid detection and dose discrimination by behaviorally resistant house flies occurs via the tarsi or the proboscis (labellum) and/or pharyngeal organs lining the esophagus.' This change is necessary to accurately reflect that the study investigates how behaviorally resistant house flies detect and discriminate among different concentrations of imidacloprid, as opposed to a specific dose.

L106 to 107 - It is recommended to include information about the duration of exposure to -20°C for the flies. This addition will enhance the precision of the experimental procedure description.

L233 to L249 - Please, it is recommended verify the text about the figures 1B and 2B. In the text, it is stated that imidacloprid concentration had a significant effect on the number of BRS flies exhibiting PER. However, upon reviewing the figures, no asterisks or symbols denoting statistical significance were observed. To ensure clarity and accurate representation, please verify that the statistical significance mentioned in the text aligns with the symbols used in the figures.

Author Response

Reviewer #1

L80 - Please, the term 'dose' should be changed to 'concentration' in the sentence 'The first is to determine whether imidacloprid detection and dose discrimination by behaviorally resistant house flies occurs via the tarsi or the proboscis (labellum) and/or pharyngeal organs lining the esophagus.' This change is necessary to accurately reflect that the study investigates how behaviorally resistant house flies detect and discriminate among different concentrations of imidacloprid, as opposed to a specific dose.

AU: we changed the expression dose with “concentration” as suggested. 

L106 to 107 - It is recommended to include information about the duration of exposure to -20°C for the flies. This addition will enhance the precision of the experimental procedure description.

AU: Exposure to -20°C varied among the flies. We considered the flies sufficiently exposed when they no longer moved and were at the bottom of the container. To be as precise as possible, we used the expression 'a few minutes' instead of 'briefly'.

L233 to L249 - Please, it is recommended verify the text about the figures 1B and 2B. In the text, it is stated that imidacloprid concentration had a significant effect on the number of BRS flies exhibiting PER. However, upon reviewing the figures, no asterisks or symbols denoting statistical significance were observed. To ensure clarity and accurate representation, please verify that the statistical significance mentioned in the text aligns with the symbols used in the figures.

AU: we added a new figure with * to indicate statistical significance.

Reviewer 2 Report

Comments and Suggestions for Authors

The science seems to be ok but sentence structure sometimes needs help. The methods for the 2 experiments are very similar and the descriptions should follow the same pattern. It would be instructive to include a fly glued to a toothpick so the orientation of the fly can be seen. It is adequately described but a photo would be great. The same items in tables and figures need to match. For example, column headings in Table 1 and 2 should be the same. Table headings and figure legends are long and wordy making them difficult to read. Much of this information could be placed in the Results text. Two paragraphs in the Discussion seem more like Introduction, and there are no comparisons to your work. These could easily be omitted. Discussions of gustatory responses of insects other than flies are interesting, but a brief mention of similarities and differences compared with house flies would be enough. And again, what does this have to do with your results? 

Author Response

Reviewer #2

It would be instructive to include a fly glued to a toothpick so the orientation of the fly can be seen. It is adequately described but a photo would be great.

AU: Photos demonstrating the fly's orientation when glued to the toothpick in addition to fly posture during the PER assays has been added.

The same items in tables and figures need to match. For example, column headings in Table 1 and 2 should be the same. Table headings and figure legends are long and wordy making them difficult to read. Much of this information could be placed in the Results text. Two paragraphs in the Discussion seem more like Introduction, and there are no comparisons to your work. These could easily be omitted. Discussions of gustatory responses of insects other than flies are interesting, but a brief mention of similarities and differences compared with house flies would be enough. And again, what does this have to do with your results? 

AU: The decision to write lengthy captions was made to aid readers who may be reviewing the work by examining the figures or tables. This allows them to quickly comprehend the methodology and approach used. While some information in the captions may have already been mentioned in the text, we have chosen to reiterate certain concepts to avoid confusion and ensure clarity, particularly as the assay are very similar.

The opening paragraphs of the discussion provide an overview of the subject and context related to gustatory receptors and behavioral response to better present our results. As there are few specific studies on this subject, we have linked the general background with our case study as much as possible. This was deemed important both for context of this study relative to previous work, particularly given that this study was performed with house flies rather than Drosophila which is a common study animal for these types of chemosensory studies.

Reviewer 3 Report

Comments and Suggestions for Authors

The manuscript by D’Arco et al. titled “Use of the Proboscis Extension Response Assay to Evaluate the Mechanism of House Fly Behavioral Resistance to Imidacloprid” is a well-written and concise report detailing the group’s work investigating the nature of house fly resistance to this commonly employed neonicotinoid insecticide.  House fly populations have exhibited behavioral resistance to baited imidacloprid formulations, this report set out to determine if the detection and dose discrimination of this insecticide is mediated by tarsal chemoreceptors or labellar/pharyngeal chemoreceptors.  The team utilized a susceptible strain of house flies and behaviorally resistant strain of house flies to test their proboscis extension response with tarsal contact only and with tarsal and proboscis contact, with three time intervals of behavioral observation for each.  The straightforward set of experiments revealed that the behaviorally resistant strain exhibited differing PER responses to low and high concentration imidacloprid/sucrose solutions at the 2 and 10 second observation times.  These results indicate that proboscis contact is required for these resistant flies to detect and discriminate among the test solutions and exhibit aversive feeding behaviors.  No significant difference in PER duration was observed in any strain when only tarsal contact was permitted. There is a fair amount of speculation in the discussion, but it is not unwarranted.

I believe that this report is of general interest and significance to insect physiologists and insect  control researchers.

The manuscript is clear and well written, and I have only the following suggestions to improve clarity:

Line 106: flies were chilled “briefly” – is there some assurance that all flies were handled in a standardized manner?

Line 131: is this concentration intended to read ug/g?  Is this a dose?  All other concentrations mentioned are by volume.

Line 221: asterisk is mentioned for figure 1, but I did not see an asterisk in my copy of the figure.

Also – it may look better if the graph titles were relegated to the figure legends.  Additionally, these titles refer to susceptible and resistant strains, but text and legends indicate specifically UCR and BRS – this should be maintained throughout.

Line 361: “Drosophila” should be italicized

Linde 364: “Blatella germanica” should be italicized

Author Response

The manuscript is clear and well written, and I have only the following suggestions to improve clarity:

Line 106: flies were chilled “briefly” – is there some assurance that all flies were handled in a standardized manner?

AU: Exposure to -20°C varied among the flies. We considered the flies sufficiently exposed when they no longer moved and were at the bottom of the container. To be as precise as possible, we used the expression 'a few minutes' instead of 'briefly'.

Line 131: is this concentration intended to read ug/g?  Is this a dose?  All other concentrations mentioned are by volume.

AU: The solution was prepared considering the volumes of water, the weight of imidacloprid and sucrose (present in the form of granules) to facilitate solution preparation. The preparation was not a dose, but a concentration expressed in grams/volumes. The term 'dose' was changed to 'concentration' to avoid confusion. We added “granular” before sucrose for clarity as to the ug/g value indicated.

Line 221: asterisk is mentioned for figure 1, but I did not see an asterisk in my copy of the figure.

AU: we added a new figure with * to indicate statistical significance.

Also – it may look better if the graph titles were relegated to the figure legends.  Additionally, these titles refer to susceptible and resistant strains, but text and legends indicate specifically UCR and BRS – this should be maintained throughout.

AU: The titles of the graphs with UCR and BRS have been revised to be as clear as possible.

Line 361: “Drosophila” should be italicized

AU: done

Linde 364: “Blatella germanica” should be italicized

AU: done

Round 2

Reviewer 2 Report

Comments and Suggestions for Authors

MS has been improved. Fly photos are great and really show what the Proboscis Extension Response looks like. Table headings still look long because table description and stats info are grouped together. Many times description is at the top and stats are at the bottom of the table. If the journal likes the tables as they are, fine with me.